Long-term dynamics of Norovirus transmission in Japan, 2005–2019

Misumi Megumi 1 2
http://orcid.org/0000-0003-0941-8537 Nishiura Hiroshi 1 3 nishiurah@gmail.com
1 Graduate School of Medicine, Hokkaido University , Sapporo, Hokkaido , Japan
2 Rumoi City Hospital , Rumoi, Hokkaido , Japan
3 School of Public Health, Kyoto University , Kyoto , Japan
Uversky Vladimir
Electronic publication date: 2021 Jul 12
Publication date: 2021
Volume: 9
Electronic Location ID: e11769
Received 2020 Dec 10; Accepted 2021 Jun 22
Copyright: © 2021 Misumi and Nishiura
Copyright year: 2021
Copyright holder: Misumi and Nishiura
License: This is an open access article distributed under the terms of the Creative Commons Attribution License, which permits unrestricted use, distribution, reproduction and adaptation in any medium and for any purpose provided that it is properly attributed. For attribution, the original author(s), title, publication source (PeerJ) and either DOI or URL of the article must be cited.
License URL: https://creativecommons.org/licenses/by/4.0/

Keywords: Caliciviridae, Natural history, Transmission, Asymptomatic ratio, Epidemiology, Epidemic

Funding: Health and Labor Sciences Research Grants 19HA1003, 20CA2024 and 20HA2007 Japan Agency for Medical Research and Development JP19fk0108104, JP20fk0108140 and JP20fk0108535s0101 Japan Society for the Promotion of Science KAKENHI 17H04701 and 21H03198 Inamori Foundation GAP Fund Program of Kyoto University Japan Science and Technology Agency CREST JPMJCR1413 SICORP (e-ASIA) JPMJSC20U3 German Federal Ministry of Health (BMG) COVID-19 Research and Development World Health Organization Hiroshi Nishiura received funding from Health and Labor Sciences Research Grants (19HA1003, 20CA2024 and 20HA2007), the Japan Agency for Medical Research and Development (JP19fk0108104, JP20fk0108140 and JP20fk0108535s0101), the Japan Society for the Promotion of Science KAKENHI (17H04701 and 21H03198), the Inamori Foundation, GAP Fund Program of Kyoto University, the Japan Science and Technology Agency CREST program (JPMJCR1413); and the SICORP (e-ASIA) program (JPMJSC20U3). This study was also supported by German Federal Ministry of Health (BMG) COVID-19 Research and Development funding to the World Health Organization. The funders had no role in study design, data collection and analysis, decision to publish, or preparation of the manuscript.

==============================
Norovirus continues to evolve, adjusting its pathogenesis and transmissibility. In the present study, we systematically collected datasets on Norovirus outbreaks in Japan from 2005 to 2019 and analyzed time-dependent changes in the asymptomatic ratio, the probability of virus detection, and the probability of infection given exposure. Reports of 1,728 outbreaks were published, and feces from all involved individuals, including those with asymptomatic infection, were tested for virus in 434 outbreaks. We found that the outbreak size did not markedly change over this period, but the variance in outbreak size increased during the winter (November–April). Assuming that natural history parameters did not vary over time, the asymptomatic ratio, the probability of virus detection, and the probability of infection given exposure were estimated to be 18.6%, 63.3% and 84.5%, respectively. However, a model with time-varying natural history parameters yielded better goodness-of-fit and suggested that the asymptomatic ratio varied by year. The asymptomatic ratio was as high as 25.8% for outbreaks caused by genotype GII.4 noroviruses. We conclude that Norovirus transmissibility has not changed markedly since 2005, and that yearly variation in the asymptomatic ratio could potentially be explained by the circulating dominant genotype.

Introduction

Norovirus is a genus of RNA viruses in the family Caliciviridae and consists of 10 genogroups (GI–GX), among which GI, GII and GIV infect humans (De Graaf, Van Beek & Koopmans, 2016; Chhabra et al., 2019). At present, GI and GII are the most frequently detected genogroups; in particular, GII.4, a genotype of the GII genogroup, continues to be most prevalent. GII.4 noroviruses evolve antigenically, producing a new variant every 2–4 years (Lindesmith et al., 2008; White, 2014). Norovirus infection is the leading cause of acute gastroenteritis worldwide (Lopman et al., 2012; Pringle et al., 2015; O’Ryan, Riera-Montes & Lopman, 2017) and is responsible for substantial morbidity, mortality, and healthcare-associated costs (Hall et al., 2013; Ahmed et al., 2014; Gaythorpe et al., 2018; Chhabra et al., 2019). Transmission is highly seasonal and most intense in the middle of winter (Phillips et al., 2010; Hall et al., 2013; Ahmed et al., 2014; Rushton et al., 2019). With improved efficiency and lower costs of the reverse transcription polymerase chain reaction (RT-PCR) technique, Norovirus infection is now known to be responsible for both sporadic infections and epidemics (Pringle et al., 2015).

Norovirus continues to evolve, modifying its pathogenesis and transmissibility (White, 2014) to ensure its continued prevalence in human populations. The major routes of transmission are via contaminated foods and the fecal–oral route (Rushton et al., 2019), although vomit from symptomatic patients also aerosolizes millions of virions. The minimum number of virions required to establish an infection is as low as 18 particles (Teunis et al., 2008). Asymptomatic carriers of Norovirus complicate the control of outbreaks (Phattanawiboon et al., 2020); the duration of viral shedding in feces and the dose of virus excreted are similar in asymptomatic carriers and symptomatic patients (Teunis et al., 2015; Newman et al., 2016). This phenomenon, especially among asymptomatic food handlers, has caused critical difficulties in controlling outbreaks (Ozawa et al., 2007; Barrabeig et al., 2010; Nicolay et al., 2011; Jeong et al., 2013; Franck et al., 2015; Sabrià et al., 2016; Chen et al., 2016; Qi et al., 2017).

Norovirus genotype GII.4 is widely prevalent worldwide (Ramani, Atmar & Estes, 2014; Kambhampati, Koopmans & Lopman, 2015). Japan is no exception, having continuously detected the genotype since 2006 (Motomura et al., 2010). Genotype GII.4 does not necessarily induce symptomatic illness in infected individuals (Ozawa et al., 2007; Miura, Matsuyama & Nishiura, 2018), increasing the possibility of asymptomatic transmission. In recent decades, this genotype has been detected most frequently, underscoring the difficulty in controlling its transmission (Miura, Matsuyama & Nishiura, 2018). The dominance of GII.4 implies that its transmissibility has increased over time (Matsuyama, Miura & Nishiura, 2017), potentially increasing outbreak sizes. Since 2007, the Food Poisoning Subgroup, Food Hygiene Committee, Ministry of Health, Labour & Welfare of Japan (2007) has issued revised guidance for food handling facilities to prevent large-scale outbreaks. The guidelines recommend routine testing of fecal samples for noroviruses among asymptomatic food handlers from October until March.

In this study, we aimed to understand long-term changes in the transmission dynamics of noroviruses and to clarify the importance of testing fecal samples of asymptomatic food handlers as part of prevention activities. To achieve the goal, we systematically collected datasets on Norovirus outbreaks documented by the prefectural institutes of public health across Japan from 2005 to 2019. In some of these datasets, both symptomatic patients and asymptomatic individuals underwent thorough laboratory-based fecal testing. We analyzed time-dependent changes in the asymptomatic ratio, the probability of virus detection, and the probability of infection given exposure in this long-term series.

Materials & methods

Data sources: epidemiological records of outbreaks

Japan consists of 47 prefectures and each prefecture has an institute of public health responsible for the laboratory testing of notifiable infectious diseases. In addition to these prefectural institutes, major urban cities (e.g., prefectural capitals) have institutes of public health; there are 83 such institutes in total throughout Japan. Whenever an outbreak of norovirus occurs, laboratory samples are analyzed and the results recorded. Once the occurrence of cases is recognized, the outbreak is detected via notifications from physicians to local health centers or through reporting by non-physicians or interviews by healthcare workers. Notification is required by Food Hygiene Law (Ministry of Health, Labour & Welfare of Japan, 2019). Fecal sampling of all asymptomatic and symptomatic individuals takes place when two or more cases arise from a food handling facility, as food poisoning is suspected. Confirmatory diagnoses are made by RT–PCR, regardless of symptoms. Feces from all individuals, including customers and food handlers, are sampled. A 10% emulsion is prepared and centrifuged at 10,000 rpm for 20 minutes in a refrigerated centrifuge, then RNA is purified from the supernatant (Food Safety Bureau, Ministry of Health, Labour & Welfare of Japan, 2003). Noroviruses are screened for genogroups GI and GII, and a subset of samples are further examined for genotyping.

The annual reports of 78 of the 83 institutes of health were screened from 2005 to 2019 (the remaining institutes do not widely publish their annual reports). In this study, we investigated the records from all these annual reports, and systematically collected datasets on the locations (prefecture) of norovirus outbreaks, the months and years of occurrence, the numbers of symptomatic cases, the numbers of exposed individuals, the numbers of laboratory tests performed and their results for symptomatic individuals, and the numbers of laboratory tests performed and their results for asymptomatic individuals (e.g., asymptomatic food handlers).

Descriptive analysis of outbreaks

We investigated the descriptive features of the outbreaks identified, including the geographic distribution, temporal distribution (by year), seasonality (measured as monthly incidence), and the distribution of facilities where the outbreaks occurred. Age was not consistently reported across outbreaks, so was excluded from the analysis. Asymptomatic cases were defined as individuals with complete absence of symptoms throughout the course of infection.

We then collected datasets on outbreak sizes over time to identify any changes in virus transmissibility and outbreak seasonality. To examine trends in outbreak size over time, two independent analyses were conducted. First, to investigate the impact of seasonal variation on virus transmission, the outbreak sizes from November to February (i.e., winter) were compared with those of outbreaks during other months. Second, yearly variations in outbreak size were investigated over time.

Estimation of the asymptomatic ratio

In addition to descriptive analyses, we used a statistical model to estimate the asymptomatic ratio (s) and other parameters including the probability that infected individuals avoided symptomatic illness (q) and the probability of virus detection in an infected individual (p), as proposed elsewhere (Miura, Matsuyama & Nishiura, 2018). While detailed mathematical descriptions can be found elsewhere (Miura, Matsuyama & Nishiura, 2018), it should be emphasized that testing fecal samples of all individuals involved in each outbreak allowed us to avoid ascertainment bias. In short, the model considered three pieces of information: the infection process, illness onset, and viral shedding. Assuming that all three parameters did not vary over time, and given the input values for the number of symptomatic individuals (ni), the number of asymptomatic individuals (mi), the number of virus-positive asymptomatic individuals (yi), and viral shedding by symptomatic individuals (xi) in each outbreak i, the total likelihood L for estimating parameters was computed as:

(1) L(p,q,s;data)=L1L2L3,

where

(2) {L1(p,s;n,m)=∏i⁡(ni+mini)(p(1−s))ni(1−p(1−s))mi,L2(p,q,s;m,y)=∏i⁡(miyi)(psq1−p+ps)yi(1−(psq1−p+ps))mi−yi.L3(q;n,x)=∏i⁡(nixi)qxi(1−q)(ni−xi).

That is, each data generation process was assumed to be described by binomial sampling. We obtained maximum likelihood estimates of p, q, and s by minimizing the negative logarithm of Eq. (1), and the 95% confidence intervals (CIs) were computed using the profile likelihood. Alternatively, we used a model in which unknown parameters varied every single year because the dominant genotype (e.g., GII.4) varied over time. To identify any improved fit despite the increased number of parameters, Akaike’s information criterion (AIC) was computed and compared across the different models.

Data sharing statement

The original outbreak size data are available as Supporting Material.

Ethical considerations

The present study was based exclusively on published data. Therefore, it did not require ethical approval.

Results

In total, the annual reports contained records for 1,728 norovirus outbreaks. Of these, 434 outbreaks had full datasets, including numbers of exposed asymptomatic individuals. Figure 1A shows the geographic distribution of cases from April 2005 to March 2019 in different geographic regions. Of the 1,728 outbreaks, the Kinki region, containing the third most populated prefecture in Japan (Osaka), had the most outbreaks (605, 35.0%). The second largest outbreak count was in the Tohoku region, located in northeastern Japan, with 393 outbreaks (22.7%). Hokkaido had only 27 documented outbreaks (1.5%). The yearly number of outbreaks was highest in 2,006, with 211 outbreaks, followed by 2010 with 171 outbreaks (Fig. 1B). The mean number of outbreaks was 118 per year. Seasonally, the highest frequency was seen in December (386 outbreaks), followed by January (311 outbreaks) and February (227 outbreaks) (Fig. 1C). The lowest frequency of outbreaks was observed in September (15 outbreaks). Figures 1D and 1E show the outbreak settings (facilities) in Nara and Tokyo by year. In Nara, approximately 70% of outbreaks occurred in nursery or elementary schools. In Tokyo, nursery schools and elderly-care facilities dominated the outbreak settings. There was no clear pattern of change in outbreak settings over time.

Figure 1 Descriptive characteristics of norovirus outbreaks in Japan in 2005–2019.

(A) Geographic distribution of outbreaks in seven regions. Numbers represent the total number of documented outbreaks in 2005–2019 (n = 1,728). (B) Long-term variations in numbers of outbreaks by year (n = 1). (C) Monthly variations in the numbers of outbreaks (n = 1). (D) and (E) Types of outbreak setting (e.g., facilities and schools). Only the data for Nara (D) and Tokyo (E) allowed us to examine the yearly numbers of outbreaks by type.

Of the 434 outbreaks in which asymptomatic individuals were tested, the most frequent locations were restaurants and food courts (203 outbreaks), followed by food-handling facilities (59 outbreaks), Japanese ryokan (lodgings; 33 outbreaks), and hotels (27 outbreaks). The sizes of confirmed outbreaks ranged from 2 to 308 cases, with a median of 16 cases (lower and upper quartiles: 9 and 30 cases, respectively). The mean number of symptomatic virus-positive cases per outbreak was 6.7 (standard deviation (SD), 5.3 individuals). The mean number of exposed asymptomatic food handlers was 7.5 individuals (SD, 7.2 individuals). The median number of asymptomatic virus-positive food handlers was 1 individual (range, 0–16 individuals).

Figure 2A compares outbreak size by season. Whereas the median (range) outbreak size was 15.5 (2–99) cases in May–October, the range of outbreak sizes expanded from November to April, with a median of 16.0 cases (range, 2–308 cases). The SD of outbreak size in summer was 20.5 individuals compared with 33.1 individuals in winter; the variance was significantly greater in winter according to an F-test (p < 0.0001). Figure 2B examines time-dependent variation in outbreak size. Overall, outbreak size remained stable over these years, with mean outbreak sizes of 5–15 cases.

Figure 2 Temporal variations in Norovirus outbreak size in Japan in 2005–2019, estimated with a virus detection program (n = 434).

(A) Seasonal variations in outbreak size. Summer is represented by May–October (n = 56) and winter by November–April (n = 378). The median outbreak sizes in summer and winter were 15.5 and 16.0, respectively. The maximum outbreak sizes in summer and winter were 99 and 308 patients, respectively, and an F-test indicated that the winter outbreaks varied more than the summer outbreaks (F = 2.6, p < 0.0001). (B) Yearly variations in the numbers of confirmed cases in 2005–2019. In both (A) and (B), the horizontal straight line represents the grand mean. The box extends from the lower to upper quartile. Whiskers extend from the 1st quartile − (1.5 × interquartile range) to the 3rd quartile + (1.5 × interquartile range).

Assuming that the asymptomatic ratio and other natural history parameters did not vary over time, the asymptomatic ratio (parameter s) was estimated as 18.6% (95% CI [17.4–19.9]), and the risks of infection and virus positivity were 63.3% (95% CI [62.1–64.5]) and 84.5% (95% CI [88.3–85.7]), respectively. Table 1 shows estimates of the natural history parameters that were assumed to change as a function of the year of outbreak. Overall, the estimates of the risk of infection and virus positivity did not show clearly increasing or decreasing trends, and the arithmetic averages for the asymptomatic ratio, risk of infection, and risk of virus positivity were 18.6%, 63.9% and 85.4%, respectively. However, we found that the asymptomatic ratio varied over time: it was smallest at 4.4% (95% CI [1.4–9.9]) in 2005, and largest at 25.5% (95% CI [21.3–30.1]) in 2007. The AIC of the model in which the natural history parameters remained constant over time was 22,292, whereas that of the model with time-varying parameters was 15,462, indicating that the model including yearly variations better captured the observed data.

Table 1 Probability of infection given exposure, the asymptomatic ratio, and the probability of virus detection estimated from outbreak data (n = 434) in 2005–2019, Japan.

Year	(p) Risk of infection (%)	(s) Risk of asymptomatic infection (%)	(q) Risk of virus-positive outcome (%)	
2005	74.3 (67.2–80.7)	4.4 (1.4–9.9)	72.5 (64.1–80.0)	
2006	64.7 (61.8–67.5)	15.9 (13.2–18.9)	79.8 (76.6–82.7)	
2007	59.8 (56.0–63.5)	25.5 (21.3–30.1)	87.1 (83.1–90.4)	
2008	68.9 (64.7–73.1)	15.7 (11.9–20.1)	82.4 (77.9–86.4)	
2009	62.4 (57.3–67.4)	17.2 (12.5–22.8)	82.0 (76.3–86.8)	
2010	65.6 (61.9–69.3)	18.7 (15.1–22.7)	88.0 (84.5–91.0)	
2011	59.1 (55.2–62.9)	19.1 (15.1–23.5)	83.6 (79.4–87.4)	
2012	61.4 (57.2–65.5)	18.7 (14.5–23.3)	83.6 (79.0–78.5)	
2013	65.7 (60.5–70.6)	18.4 (13.7–23.9)	94.1 (90.2–96.9)	
2014	59.0 (53.9–64.0)	16.6 (11.9–22.1)	91.1 (87.4–95.3)	
2015	56.7 (50.8–62.6)	25.3 (18.8–32.8)	80.7 (73.5–86.8)	
2016	71.8 (64.2–78.7)	24.3 (16.8–33.0)	92.0 (85.2–96.5)	
2017	61.8 (53.2–70.1)	21.3 (13.1–31.5)	85.5 (76.0–92.5)	
2018	68.3 (59.8–76.4)	22.2 (14.1–32.1)	81.8 (72.2–89.3)	
2019	59.0 (51.9–65.9)	15.7 (9.7–23.4)	95.8 (90.5–98.7)	
Note:

Numbers in parentheses represent 95% confidence intervals. All three parameters were assumed to vary every year.

Table 2 shows the natural history parameters by genogroup. The risks of infection and viral shedding were similar across genogroups. However, the asymptomatic ratio of GII.4 genotype outbreaks was as high as 25.8 (95% CI [22.0–29.9]). This estimate was higher than the asymptomatic ratio of GI group outbreaks alone or GII group outbreaks alone when genotyped GII.4 was excluded, but these differences were not statistically significant. More detailed comparisons of genogroup/genotype data are available as Online Table S1.

Table 2 Probability of infection given exposure, the asymptomatic ratio, and the probability of virus detection estimated by genogroup and genotype based on outbreak data for 2005–2019, Japan.

Groups	(p) Risk of infection (%)	(s) Risk of asymptomatic infection (%)	(q) Risk of virus -positive outcome (%)	
GI	62.7 (57.5–67.8)	18.6 (13.6–24.4)	82.7 (76.9–87.5)	
GII	62.7 (61.2–64.2)	18.3 (16.8–20.2)	84.4 (82.9–85.9)	
GI + GII	63.7 (60.0–67.3)	7.3 (5.0–10.3)	82.2 (78.3–85.6)	
GII.4	65.2 (61.7–68.7)	25.8 (22.0–29.9)	85.5 (81.9–88.7)	
Note:

Numbers in parentheses represent 95% confidence intervals. All three parameters were assumed to vary with genogroup or genotype. Groups were mutually exclusive (e.g., GI+GII includes outbreaks that involved both GI and GII detected and were not counted as part of the GI group or GII group). Similarly, GII represents outbreaks caused by genogroup GII, excluding those genotyped as GII.4. Note that the GII group could still include GII.4 viruses that were not genotyped.

Discussion

In this study, we investigated the long-term dynamics of norovirus outbreaks in Japan from 2005 to 2019 by systematically analyzing reports of 1,728 outbreaks published by the prefectural institutes of public health and examining the temporal and spatial distributions of the outbreaks. We showed that outbreak size did not vary markedly over the years, but the variance in outbreak size increased during winter (November–April). Assuming that natural history parameters did not vary with time, the asymptomatic ratio, the probability of virus detection, and the probability of infection given exposure were estimated to be 18.6%, 63.3% and 84.5%, respectively. The model with time-varying parameters yielded a smaller AIC value, and suggested that the asymptomatic ratio varied by year. The asymptomatic ratio for infection with genotype GII.4 was as high as 25.8%.

In this study, we demonstrated that outbreak size did not vary and that the natural history parameters governing the risk of infection and the risk of a virus-positive outcome remained stable from 2005 to 2019, indicating that there has been no apparent trend or increase in the transmissibility of Norovirus over this period. The absence of marked increases in these parameters, as was seen following the 2003/04 season by Matsuyama, Miura & Nishiura (2017), suggested that an increase in the infectiousness of the virus occurred between 2003 and 2005, shortly before the period covered by our study. Transmissibility was considered to have peaked in 2006/07 by Matsuyama, Miura & Nishiura (2017). We did not identify a similar trend in the present study, potentially in part because of the limited number of samples from 2005 and earlier. However, yearly variations in the asymptomatic ratio were identified, potentially reflecting the dominant genotype circulating in the corresponding winter; an increase in the asymptomatic ratio was particularly noticeable in 2007. However, genotyping was not complete for all genogrouped samples, and other published studies have suggested that the asymptomatic ratios of GII.4 and other genotypes were overall comparable (Bucardo et al., 2017; Saito et al., 2014; Colston et al., 2019; Bhavanam et al., 2020). Although no long-term trends were identified in our study, marked seasonal variations were observed. Dey et al. (2010) sampled Noroviruses between 1995 and 2007 across seven different geographic areas of Japan and showed that the annual peaks always occurred between November and January. Our findings echo those of Dey et al. (2010); the increased variance in outbreak size during winter (without a corresponding increase in the mean outbreak size) indicates that seasonal preference was not related to increased transmissibility but rather an increased number of outbreaks in winter season.

The large asymptomatic ratio estimated for the GII.4 genotype is consistent with the results of a prior study by Miura, Matsuyama & Nishiura (2018) in Japan. However, clinical surveillance studies have indicated that GII.4 likely caused more symptomatic infections than other genotypes (Kazama et al., 2016; Fumian et al., 2019). Miura, Matsuyama & Nishiura (2018) estimated the asymptomatic ratio to be high as 32.1% in 2006, whereas our estimate during the years examined was 25.8%. Modeling studies can avoid the ascertainment biases of laboratory testing and genotyping and allow explicit statistical estimation of the asymptomatic ratio. The basis for inconsistencies between modelling and surveillance studies should be further explored by comparing detailed clinical symptoms. Our goal here was to study the long-term dynamics of Norovirus outbreaks, whereas it was evident from the results of Miura, Matsuyama & Nishiura (2018) that the year 2006 was clearly dominated by GII.4 (Dey et al., 2010; Matsuyama, Miura & Nishiura, 2017). Therefore, we report here that the asymptomatic ratio varies by year, whereas the transmissibility of the virus in the last decade has shown no clear trend. Unfortunately, the causal impact of GII.4 in increasing transmissibility could not be investigated in the present study. In the literature, the dominance of GII.4 has not been attributed to transmissibility but rather to epochal evolution and continuous changes in antigenicity (Debbink et al., 2012; Parra et al., 2017; Tohma et al., 2019). Our results showed that there were no marked changes in the composition of genogroups/genotypes during the period of our analysis.

This study had three technical limitations. First, reported outbreaks must have been greater in size than sporadic occurrences, so our study samples may have been affected by reporting bias. For this reason, Norovirus transmissibility could have been overestimated. In fact, Qi et al. (2018) estimated that the asymptomatic ratio based on 15 outbreak datasets was high. Second, our estimates are predominantly based on the choice of food handlers as the control group, who may be more exposed to the virus than other individuals. If so, the transmissibility of Norovirus could have been overestimated because of sampling bias. Similarly, larger outbreaks would have been sampled more frequently, again elevating estimates of transmissibility. Third, our model involved simplistic assumptions (e.g., viral shedding from food handlers). Norovirus can be detected in feces for up to 4 weeks after infection (Rockx et al., 2002; Kirkwood & Streitberg, 2008), and virus excretion occurs even during the recovery period (Bucardo, 2018). Because asymptomatic hosts were all sampled among food handlers, the possibility that individuals within the recovery period were included cannot be ignored.

Although further studies are required, the present study was based on systematically collected outbreak data from 2005 to 2019, with a sample size of 1,728 outbreaks across Japan. In conclusion, our data show that the transmissibility of Norovirus has not changed over time. However, yearly variation in the asymptomatic ratio suggested a potentially higher asymptomatic ratio for GII.4 compared with other genotypes. This possibility should be further verified in future studies.

Supplemental Information

Supplemental Information 1 The original outbreak size data.

Click here for additional data file.

Supplemental Information 2 Genotype distribution data of 434 outbreaks in Japan.

Click here for additional data file.

We thank Janine Miller, PhD, from Edanz Group for editing a draft of this manuscript.

Additional Information and Declarations

Competing Interests

Author Contributions

Data Availability

Hiroshi Nishiura is an Academic Editor for PeerJ. Otherwise the authors declare that they have no competing interests.

Megumi Misumi performed the experiments, analyzed the data, prepared figures and/or tables, authored or reviewed drafts of the paper, and approved the final draft.

Hiroshi Nishiura conceived and designed the experiments, performed the experiments, analyzed the data, authored or reviewed drafts of the paper, and approved the final draft.

The following information was supplied regarding data availability:

The original outbreak size data are available in the Supplemental File.

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
