# Peer review of "Long-term dynamics of Norovirus transmission in Japan, 2005–2019"

_PeerJ, doi:10.7717/peerj.11769_

## Round 0.1 · original submission · Major Revisions

Please address all the concerns raised by the reviewers and amend your manuscript accordingly.

Reviewer 1 ·

Basic reporting

no comment

Experimental design

no comment

Validity of the findings

no comment

Additional comments

The manuscript written by Misumi and Nishiura descriptively analyzed the norovirus outbreaks reported in public sectors in Japan, 2005-2019, and built statistical models to predict the risk of infection, asymptomatic ratio, and risk of norovirus positive outcome. The outbreak data was nicely collected and summarized well, providing interesting insights on temporal dynamics of transmissibility and asymptomatic ratio. Authors could expand more the method section to explain the robustness and/or limitation of the available dataset. More specific analyses/descriptions to the outbreak settings would be helpful to gain insights on norovirus infection patterns with more biologically relevant explanations/hypotheses.


1. The authors’ hypothesis is not very clear to this reviewer. Why the dominance of GII.4 implies increase transmissibility of this virus -resulting in larger outbreak size- over time? If that is true it is easily visible from the annual epidemic curves showing the increase of the infection cases, which seems not true in Fig. 1. I think it is not necessarily increase its transmissibility but could you please explain a little bit more the hypothesis? The dominance of this genotype has been partly explained by its epochal evolution and continuous changes of its antigenicity (Debbink et al. PLoS Pathog 2012; Parra et al. PLoS Pathog 2017; Tohma et al. MBio 2019), which have probably resulted in the increase of susceptible population to the new variants. That said, it should be very interesting if authors could estimate those parameters for GII.4 and non-GII.4 viruses, parsing those into corresponding variants circulated in each time period. Also, if possible it might be interesting to look over the outbreak size, number of outbreaks, genotype distribution, and parameters by outbreak settings, e.g. nursery, elderly facility, restaurants, hotels.

2. Please describe more detail about the dataset and data collection procedures; definition of outbreaks, sampling strategies (collected fecal samples from all the symptomatic patients or a subset of them? How about asymptomatic cases? What is the criteria for diagnosing cases as symptomatic? How did they define the exposure to the viruses?), and viral detection procedure (norovirus screening detected all three human genogroups? Are they genotyped by PCR, or genomes sequenced?) are not clearly written in the materials and methods section. If genotype information is available in each outbreak, it is helpful to show the prevalent genotypes in each of them or in each year. As I believe this data was not collected in cohort studies, The ratio of asymptomatic infections and data for the model fit largely relies on the sampling strategy from asymptomatic cases in each institute or outbreak settings.

3. Although authors claimed that findings in this manuscript was consistent with the study from Matsuyama et al. (lines 188-190), it is not reflecting the sharp drop of the reproduction numbers in 2011-2012 or increase in 2013 in their study. Besides, in Matsuyama’s study the best fit model presented steep rise of the reproduction number from 2006/07 to 2007/08 season, which was included in the year-range in this study, and seems not consistent with data here. Parameters estimated were different between these studies but it would be nice if authors could explain the differences between the studies.

4. In addition, authors associated the yearly variation in the asymptomatic ratio with the change of dominant genotype circulating in the corresponding winters (lines 190-192, 223-224), without showing any data of dominant genotypes in each winter season. Again, it is helpful if authors could provide such information and conduct additional correlation analyses to prove this hypothesis. Authors commented (without statistical significance in this study and Miura’s study) that high asymptomatic ratio could be attributed to the dominance of GII.4 virus (line 224); however, multiple birth cohort studies showed higher or similar asymptomatic ratio upon GII.4 infection as compared from that of other genotypes (Ballard PLoS ONE 2015; Burcado Infect Genet Evol 2017; Saito Clin Infect Dis 2014; Colston JID 2019; Bhavanam Microorganisms 2020). Also, waste water monitoring along with clinical surveillance indicated wide range of genotypes distributed in sewage samples, suggesting that clinical samples were skewed to symptomatic infections from GII.4 while more asymptomatic/mild infections with different genotypes circulated in the community (Kazama Water Res 2016; Fumian Environ Int 2019). Please clarify any inconsistencies between the modeling analyses and those cohort/monitoring studies.

Minor comments:

5. Lines 54-55: Reference for the ’18 particles’ is incorrect. It should be Teunis et al. J Med Virol 2008.

6. Line 176: “systematically analyzing reports of 1728 outbreaks by published the prefectural institutes” should be “systematically analyzing reports of 1728 outbreaks published by the prefectural institutes”?

·

Basic reporting

The English is clear. Reference also good. The background information is adequate. However, the author did not share any information about the model.

Experimental design

There is no experiment design specifically for the paper. The author has predicted the risk of infection.

Validity of the findings

no comment

Reviewer 3 ·

Basic reporting

Long-term dynamics of Norovirus transmission in 2005 – 2019 in Japan.
Megumi Misumi, Hiroshi Nishiura.

Norovirus infections are a major cause of gastroenteritis outbreaks. Foodborne transmission is an important route for the spread of virus and globalization of the food industry enables norovirus outbreaks on an international scale. The rapid rate of genetic and antigenic evolution of noroviruses makes it difficult to develop vaccines or therapies. Thus, it is really important to determine the dynamics of transmission of noroviruses over many years and also at the level of different genogroups. In this study, Misumi et al., address the dynamics of norovirus transmission from 2005 -2019 in Japan and also examined the predominant strain that caused the majority of clinical cases. The authors investigated the annual reports of 1728 outbreaks from the data obtained from 78 institutes of health in Japan. They reported geographic distribution of outbreaks in seven regions. They found that Kinki region had the highest number of outbreaks. In addition, they examined yearly and monthly variations in number of outbreaks. With data available only from Nara and Tokyo, they also reported yearly outbreaks of norovirus by type of outbreak settings. In Figure 2, they reported the seasonal (2A) and yearly (2B) variation of number of outbreaks from 2005-2019. They found large variation in outbreaks in winter compared to months of summer. Misumi et al., also found that noroviruses of the genotype GII.4 has been a predominant strain circulating in Japan (Table 1 and 2). Their study led to an interesting conclusion that the number of outbreaks of norovirus has not changed significantly since 2005 in Japan. It will be interesting to know the transmissibility of noroviruses on an international scale because of ever increasing globalization of food industry.

Overall, it is an important and essential study reporting the changes in outbreaks of noroviruses nationally since 2005. It is a nice paper with carefully acquired data, it is well written. In my view, this manuscript is appropriate for publication in PeerJ.

Experimental design

Yes, the research question is well defined, investigated rigorously. Methods described with sufficient detail.

Validity of the findings

Their data is statistically sound and controlled. Conclusions well stated.

Additional comments

Very minor issue:

Please label both x and y axis in Figure 1D and E.

---

## Round 0.2 · Minor Revisions

Please address remaining critiques of reviewer #1 and amend your manuscript accordingly.

Reviewer 1 ·

Basic reporting

The revised version of this manuscript addressed all of the concerns and comments from this reviewer.

Experimental design

no comment

Validity of the findings

no comment

Additional comments

I have just a few minor comments on figures and tables.

1. Where are Nara and Tokyo in the map? It would be helpful if the location of those is indicated in the map in Fig.1A.

2. Is there any reason the order of each group in the graph is different between Fig.1D and 1E? I mean, in the Tokyo graph, the order of the groups in the bar plot and legend is different (oppositely ordered; Other>Eldery>Elementary>Nursery from top to bottom). Also, y-axis title is missing in Fig.1D.

3. Table 2: What does “GI+GII” mean? Co-infection/detection of GI and GII or a dataset that included all the outbreaks caused by GI or GII? If the latter is the case, why risk of asymptomatic infection is much lower than those with GI or GII alone?

·

Basic reporting

The author has addressed the comments

Experimental design

The author has addressed the comments

Validity of the findings

The author has addressed the comments

Reviewer 3 ·

Basic reporting

The authors addressed most of our concerns and made required changes to the manuscript . Thus, the manuscript is appropriate for publication in PeerJ Journal.

Experimental design

No comments.

Validity of the findings

No comments.

---

## Round 0.3 · accepted · Accept

Thank you very much for addressing the remaining concerns of the reviewer. I am really pleased to accept your revised manuscript now.